# Development and External Validation of a Nomogram Predicting Early Recurrence of Gallbladder Cancer Using Preoperatively Available Prognosticators: A Korean Multicenter Retrospective Study

**DOI:** 10.3390/cancers17091450

**Published:** 2025-04-26

**Authors:** Hyun Jeong Jeon, So Kyung Yoon, Boram Park, Hyeong Seok Kim, Hochang Chae, Hongbeom Kim, Sang Hyun Shin, In Woong Han, Jin Seok Heo, Okjoo Lee, So Jeong Yoon

**Affiliations:** 1Division of Hepatobiliary-Pancreatic Surgery, Department of Surgery, Kyungpook National University Chilgok Hospital, Kyungpook National University School of Medicine, Daegu 41404, Republic of Korea; 2Department of Surgery, Soonchunhyang University Seoul Hospital, Soonchunhyang University College of Medicine, Seoul 04401, Republic of Korea; 3Biomedical Statistics Center, Research Institute for Future Medicine, Samsung Medical Center, Seoul 06351, Republic of Korea; brpark@inha.ac.kr; 4College of Medicine, Inha University, Incheon 22332, Republic of Korea; 5Division of Hepatobiliary-Pancreatic Surgery, Department of Surgery, Samsung Medical Center, Sungkyunkwan University School of Medicine, Seoul 06351, Republic of Korea; hs0853.kim@samsung.com (H.S.K.);; 6Division of Hepatobiliary-Pancreatic Surgery, Department of Surgery, Soonchunhyang University Bucheon Hospital, Soonchunhyang University College of Medicine, Bucheon 14584, Republic of Korea

**Keywords:** gallbladder cancer, cholecystectomy, recurrence, nomogram

## Abstract

Gallbladder cancer is a rare but aggressive disease with a high likelihood of returning after surgery. Identifying patients who are at increased risk of early recurrence can help doctors make better treatment decisions before performing invasive procedures. In this study, we analyzed data from patients who underwent surgery for gallbladder cancer and identified preoperative factors that were linked to early recurrence. Based on these findings, we developed an online calculator that estimates the individual risk of early cancer return before surgery. The tool was also tested using data from other hospitals to confirm its reliability. This approach may support doctors and patients in making more personalized treatment decisions, particularly for high-risk individuals.

## 1. Introduction

Gallbladder cancer (GBC) is the most common cancer of the biliary tract, with a high prevalence in East Asia [1]. The only curative treatment for GBC is surgical resection. However, the prognosis of patients with GBC is poor even after surgical resection. In addition, GBC has a high recurrence rate after surgical resection. It has been reported that more than 20% of patients experience recurrence within a year after curative surgery, including extensive procedures, such as hepato-pancreato-duodenectomy [2].

Several prognostic factors are known to be related to survival of GBC. The most well-known predictive factor is the TNM stage of the American Joint Committee on Cancer (AJCC) [3]. Adjacent bile duct invasion, positive resection margins, and tumor differentiation have also been identified as risk factors for survival after resection of GBC [4]. However, due to the rarity of GBC, there are only limited data available regarding such risk factors. Furthermore, most prognostic factors are postoperatively confirmed pathological indicators, making it difficult to predict prognosis before high-risk surgeries.

The purpose of the present study was to investigate preoperatively available prognostic factors associated with early recurrence and survival of GBC. This study also aimed to develop a nomogram for predicting recurrence and survival of GBC in the preoperative phase using identified risk factors.

## 2. Materials and Methods

### 2.1. Patient Database

Patients who underwent curative-intent resection for GBC at Samsung Medical Center (Seoul, Republic of Korea) between 2008 and 2017 were included in the development cohort (*n* = 251). Their clinicopathologic data were retrospectively reviewed. The external validation cohort included patients who underwent surgery for GBC in three Korean tertiary centers: Soonchunhyang University Bucheon Hospital, Kyungpook National University Chilgok Hospital, and Soonchunhyang University Seoul Hospital. The date of the last follow-up was 31 December 2023. This study was approved by the Institutional Review Board (IRB) of Samsung Medical Center, Seoul, Korea (IRB approval no. 2022-10-042).

### 2.2. Perioperative Data and Definitions of Variables

Preoperative data, including laboratory and radiologic findings, were collected. Underlying chronic liver disease (CLD) was defined when there were records from hepatologists or when patients had been prescribed antiviral drugs for hepatitis. Preoperative symptoms included fever and/or abdominal pain requiring hospitalization or outpatient visits. Sarcopenia was measured using preoperative computed tomography (CT) scans. The following diagnostic cut-off values for sarcopenia were based on a cohort study from a Korean national institution [5]: skeletal muscle index (SMI = skeletal muscle area at L_3_/height_2_) < 50.18 cm^2^/m^2^ for males and SMI < 38.63 cm^2^/m^2^ for females. Sarcopenic obesity was defined as visceral fat area (VFA)/SMI ≥ 2.5, according to our previous study [6].

Clinical TNM staging was determined using preoperative CT scans, magnetic resonance imaging (MRI), and positron emission tomography (PET) scans, if available. All the patients underwent contrast-enhanced CT as the standard preoperative imaging modality. In cases where the CT findings were equivocal, MRI was additionally performed. For metastasis workup, chest CT and/or PET-CT were selectively used based on clinical indication. Imaging was reviewed by a specialized pancreatobiliary radiology team under consensus reading protocols to ensure consistency. Clinical suspicion of lymph node metastasis was defined based on one or more of the following imaging features: short-axis diameter >10 mm, rounded shape, loss of fatty hilum, or contrast enhancement pattern suggestive of malignancy.

Extended cholecystectomy included cholecystectomy combined with hepatic resection, extrahepatic bile duct resection, or hepato-pancreato-duodenectomy with regional lymph node dissection. Pathology was reported based on the 8th AJCC staging system [3].

### 2.3. Recurrence and Survival

Patients regularly visited the outpatient clinic at three- to six-month intervals after surgery for follow-up evaluations with laboratory tests, including tumor markers and abdominal pelvic CT scans. Cancer recurrence was determined based on elevated tumor markers (carcinoembryonic antigen (CEA) or carbohydrate antigen 19-9 (CA 19-9)) and suspicious lesions in CT scans during postoperative surveillance. An additional PET scan or tissue biopsy was performed to confirm recurrent tumors. Recurrence-free survival (RFS) was measured by the time from surgery to the date of recurrence or last follow-up. Early recurrence was defined as recurrence within a year from the day of surgery (RFS < 12 months). Overall survival (OS) was defined as the time between surgery and death from any cause before 31 December 2023.

### 2.4. Statistical Analysis

Binary logistic regression and Cox regression analyses were performed to identify the risk factors for early recurrence and survival. *p*-values of less than 0.05 were regarded as statistically significant. The odds ratios (OR) or hazard ratios (HR) with 95% confidence interval were reported. The optimal cut-off values of the continuous variables were obtained using the Youden Index to maximize the sum of sensitivity and specificity.

The predictive nomograms were developed using variables showing statistical significance in binary logistic regression and Cox regression analyses. Predictability was assessed using the area under the receiver operating characteristic (ROC) curve (AUC) for the logistic regression model and Harrell’s C-index for the survival model. External validation of the newly developed models was performed, and the predictive power was measured by AUC and calibration plot. All the statistical analyses were conducted using SAS software version 9.4 (SAS Institute Inc., Cary, NC, USA) and R software version 4.0.5 (The R Foundation for Statistical Computing, Vienna, Austria).

## 3. Results

Table 1 shows the demographic and clinicopathologic characteristics of the development cohort. Among 251 patients, 106 (42.2%) patients were males, and eleven (4.4%) patients had CLD. One hundred and twenty-six (50.2%) patients had preoperative symptoms. Twenty (8.0%) patients had sarcopenia, and 103 (41.0%) patients had sarcopenic obesity. Regarding the pathologic outcomes, 35 (13.9%) patients had Tis or T1 disease, 153 (61.0%) patients had T2 disease, and 63 (25.1%) patients had T3 or higher disease. Ninety-seven (38.6%) patients had lymph node metastasis, and 48 (19.1%) had early recurrence.

The results of the risk factor analysis for early recurrence using preoperative parameters are shown in Table 2. In the multivariable analysis, male sex (OR: 2.397, 95% CI: 1.069–5.375, *p* = 0.034), CLD (OR: 7.180, 95% CI: 1.317–39.134, *p* = 0.023), preoperative symptoms (OR: 4.481, 95% CI: 1.879–10.689, *p* < 0.001), elevated CEA (> 2.5 ng/mL) (OR: 2.431, 95% CI: 1.067–5.541, *p* = 0.035), sarcopenic obesity (OR: 2.366, 95% CI: 1.038–5.395, *p* = 0.041), clinically T3 or higher stage (OR: 12.375, 95% CI: 2.728–56.130, *p* = 0.001), and clinically suspicious lymph node metastasis (OR: 8.924, 95% CI: 3.375–23.597, *p* < 0.001) were significantly associated with early recurrence.

The results of the risk factor analysis for OS are shown in Table 3. Age (HR: 1.604, 95% CI: 1.259 -2.044, *p* < 0.001), male sex (HR: 1.953, 95% CI: 1.284–2.970, *p* = 0.002), CLD (HR: 5.042, 95% CI: 1.872–13.582, *p* = 0.001), preoperative symptoms (HR: 1.752, 95% CI: 1.121–2.736, *p* = 0.014), elevated CA19-9 (> 37 U/mL) (HR: 1.594, 95% CI: 1.022–2.488, *p* = 0.040), co-existing gallstones (HR: 1.709, 95% CI: 1.007–2.900, *p* = 0.046), tumor size (>2 cm) (HR: 2.273, 95% CI: 1.253–4.122, *p* = 0.007), clinically T3 or higher stage (HR: 2.952, 95% CI: 1.131–7.702, *p* = 0.027), and clinically suspicious lymph node metastasis (HR: 2.973, 95% CI: 1.829–4.834, *p* < 0.001) were independent risk factors for OS in the multivariable analysis.

Table 4 shows comparisons of the demographic and clinicopathologic characteristics between the development cohort and the validation cohort. The validation cohort consisted of a total of 176 patients. There were more patients with preoperatively elevated CEA in the validation cohort than in the development cohort (38.1% vs. 23.5%, *p* = 0.001). The number of patients with co-existing gallstones was higher in the development cohort than in the validation cohort (14.7% vs. 9.1%, *p* = 0.020). Patients in the development cohort had more advanced cT (*p* < 0.001) and cN (*p* < 0.001) stages than those in the validation cohort.

Based on the multivariable analysis using the data of the development cohort, nomograms were developed to predict early recurrence (Figure 1A) and overall survival (Figure 1B). The ROC curve (Figure 2A) and calibration plot (Figure 2B) of the nomogram for early recurrence were drawn from internal validation using 2000 bootstraps. The AUC was 0.872 (95% CI: 0.817–0.927). Figure 3 shows the results of external validation, and the AUC was 0.703 (95% CI: 0.613–0.793). The calculator is accessible online at http://gb4u.medicaldb.co.kr (accessed on 1 April 2025) (Appendix A).

## 4. Discussion

GBC remains a challenging malignancy due to its rarity and poor prognosis. Despite advances in surgical and medical oncology, the overall survival rate for GBC patients remains dismally low. Surgical resection, the mainstay of curative treatment, is typically classified into simple cholecystectomy and radical (extended) cholecystectomy. Radical cholecystectomy is recommended for T2 or more advanced disease because of its better survival rates than simple cholecystectomy. However, it is also associated with higher complication rates [7]. Even after this aggressive surgical procedure, the risk of early recurrence remains significantly high [2,8], underscoring the critical need for reliable predictive tools to support preoperative clinical decision-making.

The results of the logistic regression analysis from the developmental phase reveal that CLD was significantly associated with early recurrence and overall survival in GBC patients. This finding suggests that hepatic comorbidities could crucially impact cancer outcomes. Hepatitis B virus (HBV) and hepatitis C virus (HCV) infections are well-known risk factors for developing biliary tract cancers (BTCs), including GBC. A meta-analysis of 48 observational studies has found that both HBV and HCV infections are associated with a significantly increased risk of BTCs [9]. Another study investigating the relationship between HBV/HCV infections and BTCs has argued that this association might be due to virus-induced chronic liver inflammation and necroinflammatory processes, which can promote cell regeneration and potentially lead to carcinogenesis [10]. Future studies should identify how CLD affects not only the development but also the progression and prognosis of GBC. It is necessary to investigate whether preoperative management of CLD could lower the risk of recurrence after curative surgery of GBC.

In terms of tumor markers, the risk factor analysis showed that elevated CEA was associated with early recurrence, while CA19-9 was found to be a predictor of overall survival. CA 19-9 has been widely used for diagnosis and surveillance of GBC. Several previous studies have identified that elevated CA19-9 is associated with poorer overall survival and higher recurrence rates, highlighting its role in predicting tumor burden and metastatic potential [11,12,13]. In contrast, the prognostic value of CEA has been less consistent. While some studies have questioned its utility due to its low sensitivity (11–33%) and weaker correlation with tumor burden [14,15], others have reported associations between elevated CEA and shortened disease-free survival or early recurrence [12,16]. In our study, we used a lower cutoff value for CEA (2.5 ng/mL) compared to the 4–5 ng/mL thresholds often adopted in other cohorts. This discrepancy in cutoff levels, along with differences in study endpoints, tumor biology, and patient demographics, may help explain the divergence from previous findings. Additionally, unlike CA19-9, which can be falsely elevated in the presence of biliary obstruction or affected by Lewis antigen status [17], CEA may reflect aspects of biologic aggressiveness or micrometastatic disease not captured through conventional imaging, potentially contributing to early relapse. These findings suggest that using CEA and CA19-9 in combination may enhance preoperative risk stratification. Further prospective research is needed to validate the clinical significance of CEA and to determine the most appropriate threshold for predicting recurrence in GBC.

Sarcopenia is a factor that has been actively investigated as a potential prognosticator of diverse malignancies. Particularly, sarcopenic obesity characterized by the presence of both sarcopenia and obesity has been reported to be associated with postoperative mortality, surgical site infections, and cardiopulmonary complications in cancer patients, leading to increased hospital stays and overall costs [18,19]. In the present study, sarcopenic obesity was found to be a risk factor for early recurrence. The clinical importance of preoperative body composition assessment lies in its modifiable nature. Unlike many other preoperative factors, sarcopenic obesity can be treated by targeted interventions, such as preoperative rehabilitation and nutritional support. Preoperative screening for sarcopenic obesity and implementation of active nutritional and exercise interventions to improve muscle mass and strength can enhance postoperative recovery and potentially improve survival outcomes [20,21]. Up to now, there has been no clinical study investigating the impact of sarcopenia management on outcomes of GBC treatment. Future prospective studies combining preoperative rehabilitation with surgical management are needed to determine the clinical significance of sarcopenia and sarcopenic obesity in patients with GBC.

Gallstones are considered a relevant factor in gallbladder carcinogenesis due to their established role in promoting chronic mucosal inflammation, potentially triggering a cascade from metaplasia to dysplasia and carcinoma [22]. Prior studies have reported a positive association between gallstones and an increased risk of gallbladder cancer incidence and mortality [23,24]. A large population-based study in Japan reported that individuals with gallstones had a significantly elevated risk of GBC compared to those without gallstones [25]. In our study, gallstones were not analyzed in terms of size or number, and their relationship with recurrence remains inconclusive. Since this was a retrospective study, we could only confirm the presence or absence of gallstones based on preoperative imaging, without access to detailed characteristics, such as size, number, or composition. Future prospective studies incorporating imaging-based quantification or pathologic confirmation of gallstone characteristics will be essential to elucidate their prognostic role in GBC.

Regarding predictive platforms, there are only limited data due to the rarity of GBC. A few recent studies have presented nomograms predicting survival after curative resection of GBC [26,27,28]. The predictive ability represented by AUC ranged from 0.716 to 0.752. However, there was a great difference between the above-mentioned nomograms and our newly developed nomogram in that previous platforms used pathologic findings (histology and tumor stage), which would be available only after surgery. However, our nomogram included only preoperatively available parameters. This is obviously important because this new nomogram can be used to estimate the probability of early recurrence in the preoperative phase to enable ‘shared decision-making’ with patients. In addition, since there are several ongoing research studies on neoadjuvant therapy for BTCs [29,30], this platform could be considered as a potential indicator for neoadjuvant treatment for GBC.

This study has several limitations. First, it was a multicenter retrospective study, which might have introduced significant biases. There might be some differences in the details of preoperative preparation and postoperative surveillance among the surgeons and institutions involved. The preoperative clinical staging (T and N stages) involved high heterogeneity due to inconsistency of the imaging modalities. In addition, patients who were referred to local hospitals postoperatively were excluded because no follow-up data were available. Despite all these limitations, the present study has a number of notable strengths. To the best of our knowledge, this is the first study presenting a platform for predicting early recurrence of GBC using only preoperatively measurable factors. The nomogram can be practically used to offer information on disease status to patients and to decide proper treatment options, with a user-friendly web calculator. Particularly for patients of advanced age with significant comorbidities, clinicians can tailor therapeutic strategies more effectively, which could potentially improve outcomes and reduce the incidence of postoperative complications associated with radical surgery. The predictive ability of the nomogram was fair, with an AUC of 0.872. The AUC was 0.703 even in external validation, indicating that the nomogram would be generally applicable, particularly for the Korean population.

## 5. Conclusions

The present study presents a valuable platform for predicting early recurrence in GBC using preoperative factors. The result of external validation using multicenter data supports its applicability in clinical practice. Future prospective studies are needed to improve the predictability of this platform and validate its effect on improving patient outcomes.

## Figures and Tables

**Figure 1 cancers-17-01450-f001:**
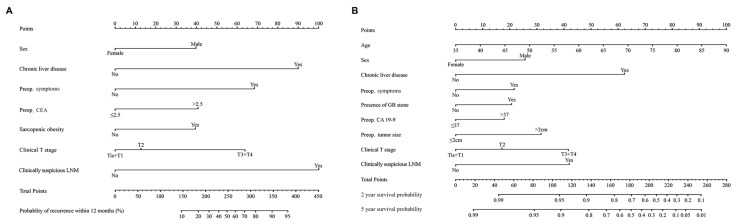
Newly developed nomograms for predicting early recurrence (**A**) and overall survival (**B**) after surgery.

**Figure 2 cancers-17-01450-f002:**
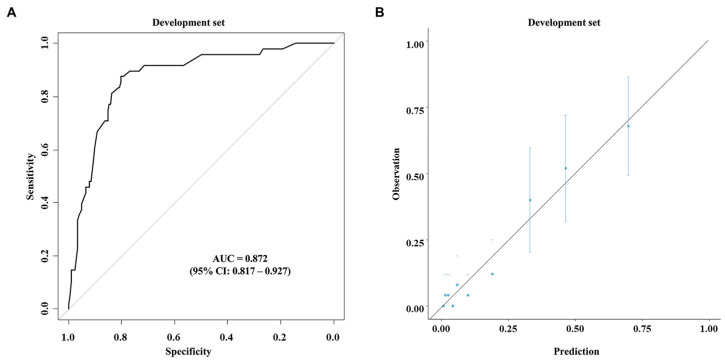
Receiver operating characteristic curve (**A**) and calibration plot (**B**) for the newly developed nomogram for predicting early recurrence.

**Figure 3 cancers-17-01450-f003:**
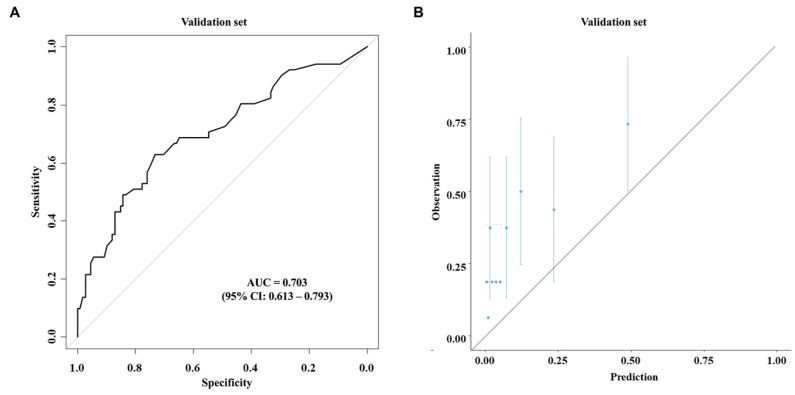
Receiver operating characteristic curve (**A**) and calibration plot (**B**) for the newly developed nomogram for predicting early recurrence using the external validation cohort.

**Table 1 cancers-17-01450-t001:** Demographic and clinicopathologic data of the development cohort (*n* = 251).

Variables	*N* (%) or Mean (±SD)	Variables	*N* (%) or Mean (±SD)
Age (years)	64.5 (±10.6)	Extended cholecystectomy	203 (80.9%)
Sex (male)	106 (42.2%)		
BMI (kg/m^2^)	23.9 (±2.8)	Pathology (AJCC 8th)	
Hypertension	104 (41.4%)	Tumor size (cm)	3.3 (±1.9)
Diabetes	34 (13.5%)	Pathologic T stage	
Chronic liver disease	11 (4.4%)	pTis + pT1	35 (13.9%)
Preop. symptoms ^1^	126 (50.2%)	pT2	153 (61.0%)
ASA score		pT3 + pT4	63 (25.1%)
I–II	242 (96.4%)	Pathologic N stage	
III–V	9 (3.6%)	pN0	154 (61.4%)
Preop. CA 19-9 (U/mL)	153.7 (±576.3)	pN1	71 (28.3%)
Preop. CA 19-9 > 37 U/mL	66 (26.3%)	pN2	26 (10.3%)
Preop. CEA (ng/mL)	2.7 (±4.7)	Differentiation	
Preop. total bilirubin (mg/dL)	0.7 (±0.7)	Well	63 (25.1%)
Preop. biliary drainage	13 (5.2%)	Moderate	115 (45.8%)
Preop. sarcopenia	20 (8.0%)	Poor	57 (22.7%)
Preop. sarcopenic obesity	103 (41.0%)	Unknown	16 (6.4%)
Preop. gallstones	37 (14.7%)		
Estimated tumor size (cm)	3.1 (±1.9)	Adjuvant treatment	61 (24.3%)
Clinical T stage			
cTis or cT1	27 (10.8%)	Early recurrence	48 (19.1%)
cT2	173 (69.0%)		
cT3 or cT4	51 (20.2%)		
Clinical N stage			
cN0	150 (59.8%)		
cN1–N2	101 (40.2%)		

^1^ Preoperative symptoms included dyspepsia, abdominal pain, fever, and jaundice requiring outpatient visits or hospitalization.

**Table 2 cancers-17-01450-t002:** Binary logistic regression analysis for early recurrence using preoperative parameters of all patients (*n* = 251).

Variables	Comparisons			Multivariable Analysis		
	Early Recurrence (−) (*n* = 203)	Early recurrence (+) (*n* = 48)	*p*	OR	95% CI	*p*
Age (years)	64.2 (±10.7)	65.8 (±10.1)	0.368			
Sex, male (ref. female)	81 (39.9%)	25 (52.1%)	0.124	2.397	1.069–5.375	0.034
BMI (kg/m^2^)	23.9 (±2.8)	23.7 (±2.93)	0.512			
Hypertension	85 (41.9%)	19 (39.6%)	0.772			
Diabetes	28 (13.8%)	6 (12.5%)	0.814			
Chronic liver disease	7 (3.4%)	4 (8.3%)	0.229	7.180	1.317–39.134	0.023
Preop. symptoms ^1^	92 (45.3%)	34 (70.8%)	0.002	4.481	1.879–10.689	<0.001
ASA scores			0.748			
I–II	195 (96.1%)	47 (97.9%)				
III–V	8 (3.9%)	1 (2.1%)				
Preop. CA 19-9	94.7 (±460.9)	402.9 (±880.3)	0.023			
Preop. CA 19-9 > 37 U/mL	45 (22.2%)	21 (43.8%)	0.023			
Preop. CEA > 2.5 ng/mL	38 (18.7%)	21 (43.8%)	0.067	2.431	1.067–5.541	0.035
Preop. total bilirubin (mg/dL)	0.7 (±0.7)	0.8 (±0.6)	0.408			
Preop. biliary drainage	9 (4.4%)	4 (8.3%)	0.280			
Preop. sarcopenia	15 (7.4%)	5 (10.4%)	0.552			
Preop. sarcopenic obesity	80 (39.4%)	23 (47.9%)	0.281	2.366	1.038–5.395	0.041
Preop. gallstones	31 (15.3%)	6 (12.5%)	0.280			
Estimated tumor size (cm)	2.8 (±1.8)	3.4 (±1.9)	<0.001			
Clinical T stage			0.021			
cTis or cT1	26 (12.8%)	1 (2.1%)		ref.		
cT2	141 (69.5%)	32 (66.7%)		2.34	0.519–10.548	0.269
cT3 or cT4	36 (17.7%)	15 (31.3%)		12.375	2.728–56.130	0.001
Clinical N stage			<0.001			
cN0	138 (68.0%)	12 (25.0%)		ref.		
cN1–cN2	65 (32.0%)	36 (75.0%)		8.924	3.375–23.597	<0.001

^1^ Preoperative symptoms included dyspepsia, abdominal pain, fever, and jaundice requiring outpatient visits or hospitalization.

**Table 3 cancers-17-01450-t003:** Cox regression model for overall survival using preoperative parameters of all patients (*n* = 251).

Variables	Univariable			Multivariable		
	HR	95% CI	*p*	HR	95% CI	*p*
Age	1.426	1.156–1.759	<0.001	1.604	1.259–2.044	<0.001
Sex, male (ref. female)	1.701	1.132–2.557	0.011	1.953	1.284–2.970	0.002
BMI	0.820	0.569–1.182	0.287			
Hypertension	1.125	0.747–1.695	0.572			
Diabetes	1.549	0.915–2.624	0.103			
Chronic liver disease	1.479	0.600–3.645	0.395	5.042	0	0.001
Preop. symptoms ^1^	1.933	1.264–2.958	0.002	1.752	0	0.014
ASA score, III–V (ref. I–II)	2.779	1.114–6.932	0.028		0	
Preop. CA 19-9 > 37 U/mL	2.362	1.555–3.585	<0.001	1.594	0	0.040
Preop. CEA > 2.5 ng/mL	2.084	1.352–3.211	<0.001			
Preop. total bilirubin	1.259	1.055–1.502	0.011			
Preop. biliary drainage	2.367	1.227–4.566	0.010			
Preop. sarcopenia	0.849	0.371–1.942	0.698			
Preop. sarcopenic obesity	0.995	0.658–1.504	0.979			
Preop. gallstones	1.498	0.895–2.507	0.124	1.709	1.007–2.900	0.046
Estimated tumor size > 2 cm	3.000	1.698–5.298	<0.001	2.273	1.253–4.122	0.007
Clinical T stage						
cTis or cT1	ref.					
cT2	1.843	0.787–4.316	0.159	1.562	0.637–3.833	0.330
cT3 or cT4	5.984	2.535–14.126	<0.001	2.952	1.131–7.702	0.027
Clinical N stage						
cN0	ref.					
cN1–cN2	3.854	2.524–5.886	<0.001	2.973	1.829–4.834	<0.001

^1^ Preoperative symptoms included dyspepsia, abdominal pain, fever, and jaundice requiring outpatient visits or hospitalization.

**Table 4 cancers-17-01450-t004:** Demographic and clinicopathologic data of the validation cohort (*n* = 176) in comparison with the development cohort (*n* = 251).

Variables	Development Cohort (*n* = 251)	Validation Cohort (*n* = 176)	*p*
Age, years	64.5 (±10.6)	67.2 (±10.6)	0.010
Sex, male	106 (42.2%)	80 (45.5%)	0.509
Chronic liver disease	11 (4.4%)	7 (4.0%)	0.838
Preop. symptoms ^1^	126 (50.2%)	84 (47.7%)	0.615
Preop. CA 19-9 > 37 U/mL	66 (26.3%)	40 (22.7%)	0.401
Preop. CEA > 2.5	59 (23.5%)	67 (38.1%)	0.001
Preop. sarcopenic obesity	103 (41.0%)	58 (32.9%)	0.357
Preop. gallstones	37 (14.7%)	13 (9.1%)	0.020
Estimated tumor size >2 cm	173 (69.0%)	120 (68.1%)	0.871
Clinical T stage			<0.001
cTis or cT1	27 (10.8%)	91 (51.7%)	
cT2	173 (69.0%)	67 (38.1%)	
cT3 or cT4	51 (20.2%)	18 (10.2%)	
Clinical N stage			<0.001
cN0	150 (59.8%)	139 (79.0%)	
cN1–N2	101 (40.2%)	37 (21.0%)	

^1^ Preoperative symptoms included dyspepsia, abdominal pain, fever, and jaundice requiring outpatient visits or hospitalization.

## Data Availability

Restrictions apply to the availability of these data. The data are not publicly available due to privacy and institutional ethical policies but may be obtained from the corresponding author, SJ Yoon, upon reasonable request and with appropriate approval.

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
