# Peer review of "Development and External Validation of a Nomogram Predicting Early Recurrence of Gallbladder Cancer Using Preoperatively Available Prognosticators: A Korean Multicenter Retrospective Study"

_cancers, 2025, doi:10.3390/cancers17091450_

Round 1
Reviewer 1 Report
Comments and Suggestions for Authors
Thank you for the opportunity to review this work. The authors took on a practical question and explained their rationale: which preoperative variables can prognosticate for the risk of recurrence and survival after resection for gallbladder malignancy. This, they explain is useful to the patients when the option of surgical resection is discussed. They based their analysis on a large number of cases (251) and used a reasonable number of variables (although the fact that only 4% of cases had chronic liver disease might cause some problems statistically- can someone check with a 'professional' statistician?)
They developed a practical model and validated it on a validation cohort. Well done.
Overall, it's s strong paper and I have only minor suggestions for its improvement:
1) include a detailed definition of "clinical suspicion of lymph node metastases"- is there not a list of measurement-based criteria that radiologists use when they make that diagnosis?
2) comment on the possibility of collinearity between "tumor>2 cm" and "stage T1/T2". If they ruled that out with a calculation, please provide that information. If they did not, why?
Author Response
Reviewer #1 comments:
Comment 0) - Thank you for the opportunity to review this work. The authors took on a practical question and explained their rationale: which preoperative variables can prognosticate for the risk of recurrence and survival after resection for gallbladder malignancy. This, they explain is useful to the patients when the option of surgical resection is discussed. They based their analysis on a large number of cases (251) and used a reasonable number of variables (although the fact that only 4% of cases had chronic liver disease might cause some problems statistically- can someone check with a 'professional' statistician?)
They developed a practical model and validated it on a validation cohort. Well done.
Response: Thank you for the insightful comment. We would like to clarify that Boram Park, a co-author of this study, is a Ph.D. in biostatistics and served as the lead statistician for all analyses presented in the manuscript.
Overall, it's s strong paper and I have only minor suggestions for its improvement:
Comment 1) include a detailed definition of "clinical suspicion of lymph node metastases"- is there not a list of measurement-based criteria that radiologists use when they make that diagnosis?
Response: We agree that clarification of the definition for “clinical suspicion of lymph node metastases” is important.
We have now added a more detailed explanation of this variable in the Methods section (2.2). Specifically, clinical suspicion was based on radiologic criteria used in our institution, including lymph nodes with short-axis diameter >10 mm, rounded shape, loss of fatty hilum, and/or enhancement patterns suggestive of malignancy on CT or MRI.
This clarification has been added to the revised manuscript in methods section as follows:
Clinical suspicion of lymph node metastasis was defined based on one or more of the following imaging features: short-axis diameter >10 mm, rounded shape, loss of fatty hilum, or contrast enhancement pattern suggestive of malignancy.
Comment 2) comment on the possibility of collinearity between "tumor>2 cm" and "stage T1/T2". If they ruled that out with a calculation, please provide that information. If they did not, why?
Response: Thank you for this important comment. We acknowledge that collinearity between “tumor size >2 cm” and “clinical T stage” may exist, given the biological and clinical correlation between tumor size and depth of invasion.
However, in our study, these two variables were treated as distinct clinical parameters, as tumor size reflects dimensional characteristics, while clinical T stage reflects tumor invasion depth, evaluated via imaging modalities such as MRI and CT. In our institution, clinical T stage is determined by the radiologic assessment of tumor extension into adjacent structures, which is not solely dependent on size.
While we did not perform a formal multicollinearity analysis in this study, we appreciate your suggestion and will consider including statistical measures such as Variance Inflation Factor (VIF) in future research to further validate the independence of predictor variables.
Also, regarding the clinical staging including tumor size and TNM staging, we have added additional explanation in Method section, as you requested in the first comment. Thank you.
Reviewer 2 Report
Comments and Suggestions for Authors
The topic of this manuscript is highly relevant and interesting, particularly given the clinical challenges associated with predicting early recurrence in gallbladder cancer. The authors are to be commended for their effort in conducting a multicenter retrospective study and attempting to build a nomogram based on preoperative parameters. The idea of incorporating easily available clinical and laboratory predictors to guide clinical decision-making is laudable.
However, the manuscript requires major revisions before it can be considered for publication. I would kindly ask the authors to address the following concerns and suggestions in their revised submission:
-
Clinically Suspicious Lymph Node Metastasis:
The authors refer to “clinically suspicious lymph node metastasis” as one of the predictors. However, the criteria for defining and evaluating this variable are not clearly described.-
What imaging modality was used (CT, MRI, PET-CT)?
-
What specific features (e.g., size, shape, enhancement pattern) were considered indicative of metastatic involvement?
-
Was there any interobserver agreement or standardization across centers?
-
-
Gallstones and Their Impact:
The presence of gallstones is briefly mentioned, but the authors do not present any detailed analysis or results related to this variable.-
Were gallstones a significant variable in the cohort?
-
Is there any observed association between gallstone size (e.g., large stones vs. microlithiasis) and recurrence?
-
Given the known pathophysiological implications of gallstones in carcinogenesis, this aspect deserves a more in-depth discussion.
-
-
Clotting Profile and Coagulation Abnormalities:
Coagulation status can be a critical factor, especially in cancer patients.-
Did the authors collect data on clotting profiles (e.g., PT, aPTT, INR, fibrinogen levels)?
-
Were there any coagulation abnormalities in the study population?
-
If so, do these correlate with recurrence or disease burden in any way?
-
-
Tumor Markers – CA19-9 vs. CEA:
The authors mention that CEA is not as strongly correlated with tumor burden as CA19-9, which differs from their own results.-
This discrepancy should be addressed more thoroughly.
-
Could the authors provide a hypothesis or explanation for this unexpected finding?
-
Are there differences in the population studied or methods used that could explain this contrast with existing literature?
-
-
References and Literature Review:
The reference list is currently too limited and lacks several key studies published in recent years.-
The authors are encouraged to incorporate more up-to-date and relevant references, especially studies on nomograms, tumor markers, and recurrence predictors in gallbladder cancer.
-
Expanding the literature review would strengthen the background and discussion sections significantly.
I look forward to reviewing a revised version of this manuscript that addresses these points in detail.
Best regards.
-
Author Response
Reviewer #2 comments:
Comment 0)
The topic of this manuscript is highly relevant and interesting, particularly given the clinical challenges associated with predicting early recurrence in gallbladder cancer. The authors are to be commended for their effort in conducting a multicenter retrospective study and attempting to build a nomogram based on preoperative parameters. The idea of incorporating easily available clinical and laboratory predictors to guide clinical decision-making is laudable.
However, the manuscript requires major revisions before it can be considered for publication. I would kindly ask the authors to address the following concerns and suggestions in their revised submission:
Response: Thank you for your valuable review. We have made our best efforts to address each point thoroughly in a point-by-point manner. We sincerely appreciate your time and consideration.
Comment 1) Clinically Suspicious Lymph Node Metastasis:
The authors refer to “clinically suspicious lymph node metastasis” as one of the predictors. However, the criteria for defining and evaluating this variable are not clearly described.
What imaging modality was used (CT, MRI, PET-CT)?
What specific features (e.g., size, shape, enhancement pattern) were considered indicative of metastatic involvement?
Was there any interobserver agreement or standardization across centers?
Response: Thank you for this important comment. We have clarified the criteria used to define “clinically suspicious lymph node metastasis.”
In our study, contrast-enhanced CT was used for all patients as the primary modality for preoperative evaluation. MRI was additionally performed in cases where CT findings were inconclusive. For metastasis workup, chest CT and/or PET-CT were performed depending on institutional protocol and patient condition.
Suspicious lymph nodes were identified based on the following radiologic features:
– Short-axis diameter >10 mm,
– Rounded morphology,
– Loss of fatty hilum,
– Strong or heterogeneous contrast enhancement.
At our center, all imaging studies were interpreted by a dedicated hepatobiliary radiology team. Lymph node status was determined via formal consensus, which minimizes interobserver variability and ensures diagnostic consistency.
These clarifications have now been added to the revised manuscript in Section 2.2., as follows:
All patients underwent contrast-enhanced CT as the standard preoperative imaging modality. In cases where CT findings were equivocal, MRI was additionally performed. For metastasis workup, chest CT and/or PET-CT were selectively used based on clinical indication. Imaging was reviewed by a specialized pancreatobiliary radiology team under consensus reading protocols to ensure consistency. Clinical suspicion of lymph node metastasis was defined based on one or more of the following imaging features: short-axis diameter >10 mm, rounded shape, loss of fatty hilum, or contrast enhancement pattern suggestive of malignancy.
Comment 2) Gallstones and Their Impact:
The presence of gallstones is briefly mentioned, but the authors do not present any detailed analysis or results related to this variable.
Were gallstones a significant variable in the cohort?
Is there any observed association between gallstone size (e.g., large stones vs. microlithiasis) and recurrence?
Given the known pathophysiological implications of gallstones in carcinogenesis, this aspect deserves a more in-depth discussion.
Response: Thank you for this important observation. We agree that a more in-depth discussion regarding the pathophysiological relationship between gallstones and gallbladder cancer is warranted.
Gallstones are indeed a significant factor, as they are known to promote chronic mucosal inflammation in the gallbladder, which may lead to metaplasia, dysplasia, and ultimately carcinoma through well-described mechanisms.
However, our study did not analyze gallstone characteristics in detail, such as stone size (e.g., large stones vs. microlithiasis), nor did it explore the direct relationship between gallstones and recurrence outcomes.
We acknowledge this as a limitation and have reflected this point in the revised Discussion section. Future studies with targeted imaging or pathologic stone characterization would be valuable to further investigate this potential association.
Added in Discussion:
Gallstones are considered a relevant factor in gallbladder carcinogenesis due to their established role in promoting chronic mucosal inflammation, potentially triggering a cascade from metaplasia to dysplasia and carcinoma. Prior studies have reported a positive association between gallstones and an increased risk of gallbladder cancer incidence and mortality. A large population-based study in Japan reported that individuals with gallstones had a significantly elevated risk of GBC compared to those without gallstones. In our study, gallstones were not analyzed in terms of size or number, and their relationship with recurrence remains inconclusive. Since this was a retrospective study, we could only confirm the presence or absence of gallstones based on preoperative imaging, without access to detailed characteristics such as size, number, or composition. Future prospective studies incorporating imaging-based quantification or pathologic confirmation of gallstone characteristics will be essential to elucidate their prognostic role in gallbladder cancer.
Comment 3) Clotting Profile and Coagulation Abnormalities:
Coagulation status can be a critical factor, especially in cancer patients.
Did the authors collect data on clotting profiles (e.g., PT, aPTT, INR, fibrinogen levels)?
Were there any coagulation abnormalities in the study population?
If so, do these correlate with recurrence or disease burden in any way?
Response: Thank you for this thoughtful comment.
We agree that coagulation abnormalities may be clinically relevant in cancer patients. In our study, however, all enrolled patients underwent curative-intent surgery, and those with clinically significant coagulopathy were deemed ineligible for surgery under the institutional anesthesia and perioperative protocols.
Therefore, although coagulation profiles (e.g., PT, aPTT, INR) were not formally collected for analysis, we believe that such abnormalities were effectively excluded from our study population.
To further investigate the prognostic value of coagulation parameters, we will consider including clotting profile variables in future prospective studies. We appreciate your helpful suggestion.
Comment 4) Tumor Markers – CA19-9 vs. CEA:
The authors mention that CEA is not as strongly correlated with tumor burden as CA19-9, which differs from their own results.
This discrepancy should be addressed more thoroughly.
Could the authors provide a hypothesis or explanation for this unexpected finding?
Are there differences in the population studied or methods used that could explain this contrast with existing literature?
Response: Thank you for pointing this out. We agree with the reviewer’s observation and have revised the Discussion section to address this discrepancy in greater detail.
Specifically, we have included literature describing the inconsistent prognostic value of CEA, and noted that our study used a lower cutoff value (2.5 ng/mL) than those commonly reported (4–5 ng/mL). We also discussed how differences in study endpoints, tumor biology, and patient characteristics may contribute to the variability in findings across studies.
Furthermore, we proposed a possible explanation that CEA may reflect subtle biologic aggressiveness or micrometastatic spread not detectable through conventional imaging. These additions aim to provide a more nuanced interpretation of our results and their clinical implications.
Added in Discussion:
In contrast, the prognostic value of CEA has been less consistent. While some studies have questioned its utility due to its low sensitivity (11–33%) and weaker correlation with tumor burden, others have reported associations between elevated CEA and shortened disease-free survival or early recurrence. In our study, we used a lower cutoff value for CEA (2.5 ng/mL) compared to the 4–5 ng/mL thresholds often adopted in other cohorts. This discrepancy in cutoff levels, along with differences in study endpoints, tumor biology, and patient demographics, may help explain the divergence from previous findings. Additionally, unlike CA19-9—which can be falsely elevated in the presence of biliary obstruction or affected by Lewis antigen status—CEA may reflect aspects of biologic aggressiveness or micrometastatic disease not captured through conventional imaging, potentially contributing to early relapse. These findings suggest that using CEA and CA19-9 in combination may enhance preoperative risk stratification. Further prospective research is needed to validate the clinical significance of CEA and to determine the most appropriate threshold for predicting recurrence in gallbladder cancer.
Comment 5) References and Literature Review:
The reference list is currently too limited and lacks several key studies published in recent years.
The authors are encouraged to incorporate more up-to-date and relevant references, especially studies on nomograms, tumor markers, and recurrence predictors in gallbladder cancer.
Expanding the literature review would strengthen the background and discussion sections significantly.
I look forward to reviewing a revised version of this manuscript that addresses these points in detail.
Best regards.
Response: Thank you very much for this constructive suggestion.
We fully agree that integrating recent and relevant literature is essential to strengthen the scientific background of the study. Gallbladder cancer, however, remains a relatively rare malignancy globally, and high-quality, recent studies—particularly those involving validated nomograms or prognosis predictors—are still limited in number.
Nevertheless, in response to your valuable comment, we have re-examined the literature specifically related to gallstones, tumor markers (e.g., CEA and CA 19-9), as these were areas explicitly highlighted in your review. We believe these additions enhance the scientific relevance and clarity of our interpretation. Thank you.
Round 2
Reviewer 2 Report
Comments and Suggestions for Authors
The authors have improved their manuscript.